# Determination of Processing Precision of Hole in Industrial Plastic Materials

**DOI:** 10.3390/polym15020347

**Published:** 2023-01-09

**Authors:** Sandor Ravai-Nagy, Alina Bianca Pop, Aurel Mihail Titu

**Affiliations:** 1Department of Engineering and Technology Management, Faculty of Engineering, Technical University of Cluj-Napoca, Northern University Centre of Baia Mare, 62A, Victor Babes Street, 430083 Baia Mare, Romania; 2Industrial Engineering and Management Department, Faculty of Engineering, “Lucian Blaga” University of Sibiu, 10 Victoriei Street, 550024 Sibiu, Romania

**Keywords:** drilling, machining precision, hole diameter, plastic materials, design of experiment, cutting parameters

## Abstract

The precision of the thread processed by tapping is closely related to the precision of the pre-drilling of the blank. Currently, the technologies for processing threads with the tap in the case of metals are well established. In this sense, there are tables and clear recommendations about the tool pairs—helical drill-tap, depending on the size of the thread, but in the case of plastics, no correlations or recommendations have been found. A well-known aspect concerns the fact that the hole made in plastics is obtained with a smaller diameter than the diameter of the drill bit used. To determine the differences between the drill bit and the diameter of the resulting hole, and its precision on different types of plastic materials, experimental research was started. At the same time, the tolerance of the resulting hole was checked and the influence of the cutting regime on the processing precision was studied. During the experiments, plastic materials often used in the aeronautical and car-building industries were used: POM-C, PA6, PEHD1000, Sika Block 700, Sika Block 960, and Sika Block 980. Following the experiments, differences in the diameter of the holes processed were found according to the plastic mass of even 0.3 mm, which is 4.4% of the diameter of the hole. Based on the experimental results and the design of the experiment, recommendations could be made about the diameter of the drill to use to obtain the desired diameter of the hole after processing.

## 1. Introduction

From a production standpoint, the thread being machined will be directly impacted by the attained hole diameter and tolerance.

In specialized literature, several studies addressing this subject have been located.

Among the most relevant is the research of [1], which studied aramid fiber-reinforced plastic. This material is used in bulletproof and armored structures. The difficulty in processing this material arises due to the high-strength aramid fibers with ductile fracture, which is different from carbon fiber. In this context, the drilling quality is affected due to delamination and burrs. The authors set up two-dimensional cutting parameters for the analysis of fiber deformation and material interface cracking. They found that, according to the model, reducing the thrust force and the radial force of the edge on the fibers is an effective way to reduce fiber deformation, and they proposed a three-point drill. In this context, they performed comparative experiments between twist drilling, “candle core drilling”, and step drilling at three points under three drilling parameters. They found that the three-point step drill changed the traditional cutting behavior of the drill exit material into a composite process.

Similar research is in [2], which approached the problem by conducting systematic experiments to evaluate cutting performance and hole quality. Upon entering the holes, the cutting edges of the twist drill peel and tear the uncut material, resulting in severe fuzzing damage. Due to the radial angles of the milling tool and the drill, the radial part of the cutting force can prestress the aramid fibers before they are cut, which effectively reduces the fuzz defect. At the exit of the hole, the extruding action of the chisel edge and severe chip adhesion are the main causes of exit damage for the twist drill and the burr tool, respectively. Due to the decrease in the thrust force and the improvement of the shearing action, the best quality of the hole is obtained by the drill. To further improve hole quality, an auxiliary approach is introduced that uses collars to effectively restrict damage by improving the bond interface strength. In the works [3,4,5], related topics are addressed on the same types of materials. Ref. [3] stated that more studies need to be done in the future for any NFRP composite cut by AWJM or LBM, and these studies should include a satisfactory range of material thicknesses and an adequate range of different input processing parameters to assess the desired outcomes and determine the most appropriate process to reduce NFRP. Milling operation has frequently been an important machining method, by which the desired dimensions and tolerances can be achieved for plate-shaped parts. In this context, ref. [4] found that the cutting force, deformation factor, and surface roughness were influenced by the feed rate and cutting speeds. In addition, increasing the number of flutes of the cutting tools reduced the cutting force, delamination factor, and surface roughness. According to [5], literature reviews indicate that only a few research articles have investigated the drilling of natural-fiber-reinforced composites. In this context, ref. [5] examined the drilling of woven jute-fiber-reinforced polymer composites in order to evaluate cutting parameters and the effect of drill types on thrust force, delamination size, and surface roughness in the drilling of jute-fiber-reinforced polymer composites, using an experimental design based on the full factorial technique. The experimental results indicated that the most significant effect was that the drill type influenced both the delamination factor and the surface roughness. All these articles [1,2,3,4,5] provide comprehensive and available information on the performance of tools for drilling aramid fiber-reinforced plastics, which can help guide process optimizations to achieve the desired hole quality.

In the case of metals, there are unanimously accepted recommendations about the diameter of the drill bit (respective of the hole to be processed). This data can be found tabularly according to the thread and its pitch in the specialized literature and in the tool catalogs that are the basis of these aspects.

Polini and Corrado [6] argued that geometric accuracy is a critical performance factor for machining, especially when one of the basic requirements is high accuracy. In their research, they present a general and systematic approach, with the aim of modeling the geometric deviation of the hole from the nominal due to the geometric errors of three essential elements of the drilling process: the locators, the workpiece reference surface, and the machine tool. In this context, they calculated the deviation from the nominal of the batches of holes on a plate to evaluate the influence of the location errors, the shape deviation of the reference surfaces of the part, and the volumetric error of the machine tool. They concluded that the volumetric error of the machine tool mainly influenced the deviation of the hole location from the nominal. Similar research in this regard is presented in papers [7,8,9,10]. Ref. [7] aimed to improve the accuracy of the hole position, optimize for stiffness, propose a robotic drilling system before processing, and compensate for the hole position error in processing.

In [8,9], the problem of the correct use of fasteners is highlighted. In this regard, the main objective of this research is to systematically review the field of fastener design. This study was useful in choosing the right fixation device, according to our research.

The study [10] provides practical guidance on drilling and drilling technology and its application in composite materials and structures. In this regard, guidance is provided with the aim of identifying the lack of knowledge in the operations of making holes in composites and highlighting the shortcomings of the current methods. The study documents the latest research, providing a better understanding of the pattern and characterization of holes produced by various technologies in composite materials.

Altan [11] analyzed burr formation and surface roughness in drilling engineering plastics. The conclusions drawn are directed towards the analysis of a relationship between the burr height and the type of chips, as well as the observation of the process variables that lead to the increase of the burr height, in turn leading to the generation of irregular and highly deformed chips.

In the paper [12], hole accuracy is analyzed in the context of processing SIKA Block M960, PA6oil, PA6SA. Similarly, in [13], surface roughness at low temperatures in the processing of HDPE1000 is analyzed.

In [14], the responses to damage induced by processing during the abrasive water jet drilling process of polymer/hybrid composites reinforced with advanced durable biofiber are analyzed. Similarly, the same research topic is also addressed in [15,16,17]. From these papers, in which processing by injection molding and water absorption were studied, it emerges that the dimensional changes of the processed piece and the consequent choice of processing dimensions must be considered.

Diaz in [18] performs an experimental analysis based on the induced drilling process, analyzing damage in bio composites. The study found that the extent of damage depends on the type of fiber and the geometry of the hole.

Other significant research in the mentioned directions belongs to [19], which studies the machinability of plastic materials reinforced with glass fiber, carbon, and aramid in drilling and secondary drilling operations. Those plastic materials are analyzed by [20,21,22,23].

In ref. [20,21], it is mentioned how composite materials are increasingly used in the aerospace, automotive, aeronautical, and marine sectors due to their high strength-to-weight ratio. Traditional machining methods have problems when machining composite materials. This paper provides a review of the findings of the literature and addresses the issues that need to be discussed.

The research [22,23] aims to study the effects of drilling factors on thermal-mechanical properties and delamination experimentally during the drilling of glass-fiber-reinforced polymer. The investigation of thermal and mechanical behaviors of the woven GRP composite laminated under different drilling bits has not been addressed comprehensively to the author’s knowledge and the literature. Therefore, the current article aims to cover this point.

The necessity and topicality of the theme represents the needs of the companies that machine the materials studied in the work at the time when the technologist must establish the tooling. Through the information provided, machining technologies can ensure machining precision through the correct choice of cutting tool and cutting parameters, which reduces the number of rejects, test times, and unnecessary tools bought.

Through this research paper, we created a knowledge matrix related to the probable diameter of the hole obtained, according to the axes: material, cutting regime, cutting tool diameter.

An approach of this systemic type has not been found in the specialized literature, which is an identified knowledge gap. A large-scale study has not been conducted to create useful tools (based on knowledge) for those who must make parts from industrial plastics with holes, i.e., internal threads.

As a result of the studies, the differences between the materials can be seen in terms of processing precision, as well as the actual diameter of the hole after processing them with the same cutting tool-drill.

To fulfill the knowledge gap, the aim of this scientific paper is:Finding the influence of the material on the precision of the processed hole;Finding the influence of the material on the diameter of the processed hole;Finding the influence of the cutting regime on the precision of the hole, with concrete reference to the deviation from cylindricity, i.e., the resulting diameter after processing.

The proposed aim will be fulfilled based on experimental research, whose variables will be defined further.

## 2. Experimental Details

### 2.1. Material Type

The semi-finished product set up for performing the experiments was used to prepare 12 samples. The size of each test piece is 50 mm × 50 mm and 16 mm thick. The total samples accumulate 12 samples × 6 types of material, resulting in 72 pieces. In each sample, 4 holes are made with the same drill.

In Table 1, according to the Quality Certificates issued when buying the semi-finished products from which the samples were made, the mechanical characteristics of the following 6 types of plastic materials are presented:Material 1: POM-C (Polyacetal-Copolymer, EN ISO 1043);Material 2: PEHD 1000 (High Density Polyethylene);Material 3: PA6 (Polyamide, EN ISO 1874-1);Material 4: SIKA BLOCK M960 (Polyurethane-Hight Density);Material 5: SIKA BLOCK M980 (Polyurethane-Hight Density);Material 6: SIKA BLOCK M700 (Polyurethane-Medium Density).

### 2.2. Processing of Holes in Test Pieces

The chipping operation used in the processing is that of drills.

The cutting tool used in the processing consists of a twist drill with a diameter of ϕ6.7. The coding of the drill is GARANT 114550 HSS-CO8. The drill characteristics are tool material: HSS Co 8, Coating TiAlN, point angle: 135°, helix angle: 35°, plain shank, number of cutting edges z = 2, point geometry shape C, hole diameter nominal tolerance h8 in case of stainless steel.

After measuring in the tool holder, the diameter of the drill bit is ϕ6.824. The CNC machining center is a Challenger 2418 CNC machine (Figure 1). In Table 2, the characteristics of the Challenger 2418 CNC machine are presented.

The cutting tool measuring machine used is the VIO 210 MICROSET, manufactured by DMG (Figure 2), and the measuring software is DMG Microvision III.

The characteristics of the cutting tool measuring machine VIO 210 MICROSET assume a spindle runout of 2 µm and repeatability ±2 µm.

The holes are measured with the CMM coordinate measuring machine type LK Metrology ALTERA S.7.5.5. The machine is equipped with a RENISHAW PH10M measuring head and CAMIO measuring software. The features are:-Volumetric accuracy: 1.8 µm + L/400 (L is the measured length).-Repeatability: 1.7 µm.-Velocity: 762 mm/s.-Acceleration: 2.306 mm/s^2^.

The variable cutting parameters chosen in conducting the research, as well as their variation levels, are:Cutting speed—3 levels: 12, 25, 37 [m/min].Feed—4 levels: 0.1; 0.2; 0.3; 0.4 [mm/rpm].

## 3. Conduct and Analyze Our Experimental Design

Regarding the planning of experiments, according to Montgomery [24], there are three basic principles, namely:Principle of randomness, based on which statistical methods require that the observations (or errors) have a random character, that is, they are randomly distributed (randomized) with respect to the parameters. Randomization of observations makes this assumption valid.The principle of replication, which involves repeating the experiment 3 to 7 times for each set of values of the input parameters. This procedure is necessary to decide the constancy of the measurements.The principle of working in “blocks”, which is used to improve the precision of the comparison between the factors used.

In the present case, to obtain the most accurate results, a four-fold replication of the experiments will be carried out. In other words, for each set of parameters, four holes will be made, for which seven measurements will be made for each hole. The reason for choosing to replicate four times is constituted by the architecture of the specimen.

By using the device and the architecture of the specimen, the locating bases used to manufacture will be identical to the measurement datum. Figure 3 shows the sample with the four holes and the fixing device on the CMM table (CMM-Coordinate Measuring Machine).

In the current research, we will work with 12 blocks. Each block corresponds to a cutting regime. This results in a total of 12 experiments for each individual material; in other words, we will have 12 blocks × 6 materials = 72 samples. Each experiment is repeated four times, as stated above, with arguments related to the reason for replication.

Following the design of experiments, the research course was set up using the factorial experiment method. The application used for the research and data processing is STATGRAPHICS. In this context, the achievement of the two previously defined aims is pursued.

Data processing was possible after obtaining the measurements, tracking the measured elements cylinder diameter and cylindricity, represented in Figure 4. The CMM calculates the diameter of the hole and its cylindricity based on two circles felt in two different planes, P1 and P2, with P0 noted as the device seating surface (Figure 3 and Figure 4).

In the following, based on the establishment of the two major research directions, the two studies will be presented. The first of them is aimed at deciding the influence of the material on the diameter of the machined hole.

## 4. Determination of the Influence of the Material on the Precision of the Machined Hole

Figure 5 and Figure 6 show a violin plot for samples that combines box-and-whisker plots with nonparametric density traces. The first step in constructing a box-and-whisker plot is to first find the median (Q2), the lower quartile (Q1), and the upper quartile (Q3) of a given set of data. We have chosen this approach because it is a useful method for comparing the distribution of several samples of quantitative data. By here results the form of the probability density function for the circularity from which the data comes. From Figure 5, of the four holes processed under the same cutting conditions, the deviations from circularity related to the hole d on the measurement plane P1 of the specimen show different variations compared to the other three holes. The same can be said about the deviation from circularity measured on hole c on the second face of the specimen (Figure 6). Based on the 2 graphs, it is found that there are 4 holes, out of the total of 288, that present anomalies. These, in our opinion, are the result of production mistakes. After the drills, there may be burrs that were not removed by deburring processes to preserve the integrity of the surface obtained by the drills. The anomalies found following the interpretation of the graphs in Figure 5 and Figure 6 are presented in Table 3 and Figure 7.

The findings reached above are presented in Figure 7, which was created using experimental data. It demonstrates the dispersion of the measured values of the circularity deviation, according to the material of the samples.

Next, a comparison of Alternative Models was made in Statgraphics based on the predefined methods in the software, which is presented in Table 4.

Table 4 shows the results of fitting several curvilinear models to the data in the case of the two measurement planes P1 and P2. Of the models fitted, the logarithmic-Y square root-X model yields the highest R-Squared value, with 91.9338% in the P1 case and 93.78% in the P2 case. Now we will conduct the analysis.

Figure 8 and Figure 9 show the fitted model plot and residual plot for the mentioned model’s circularity in P1 and P2, respectively.

In the P1 and P2 situations, the dependent variable is Circularity 1 and Circularity 2, respectively, and the independent variable is Material.

The Logarithmic-Y square root-X model is:Y = exp(b × sqrt(X))(1)

The obtained data resulting from the analyses is presented below (Table 5 and Table 6).

The output shows the results of fitting a logarithmic-Y square root-X model to describe the relationship between Circularity 1 and Circularity 2, respectively, and Material. The equations of the fitted model are:Circularity 1 = exp(−2.02811 × sqrt(Material))(2)
Circularity 2 = exp(−1.98681 × sqrt(Material))(3)

Since the *p*-value in the ANOVA table is less than 0.05, in both situations, there is a statistically significant relationship between Circularity 1 and Circularity 2, respectively, and Material at the 95.0% confidence level.

The R-Squared statistic shows that the model as fitted explains 91.9338% of the variability in Circularity 1 and 93.7764% of the variability in Circularity 2. The correlation coefficient equals −0.958821 for Circularity 1 and −0.968382 for Circularity 2, showing a relatively strong relationship between the variables. The standard error of the estimate shows the standard deviation of the residuals to be 1.13177 for Circularity 1 and 0.964281 for Circularity 2. The mean absolute error (MAE) of 0.889192 is the average value of the residuals in the Circularity 1 case and 0.680314 in the Circularity 2 case.

As a conclusion, based on the statistical data obtained, 4 values were identified out of a total of 288, which represents a percentage of 1.38% of outliers. From a manufacturing perspective, 1.38% of a batch is an acceptable percentage.

## 5. Determination of the Influence of the Material on the Diameter of the Hole Machined

After analyzing the measured and centralized data in Table 7 and Table 8, we find that the diameter of the hole is smaller than the diameter of the tool used. The elasticity of the material being processed and the cutting regime utilized in the processing have an impact on the discrepancy between the diameter of the hole and the cutting tool.

**Table 7 polymers-15-00347-t007:** The measured diameters of the holes depending on the material and cutting parameters.

Cutting Parameter No.	1	2	3	4	5	6	7	8	9	10	11	12	(Figure 10)		
v [m/min]	12	25	37	12	25	37	12	25	37	12	25	37	Hole diameter		Δ diameter
s [mm/tooth]	0.05	0.05	0.05	0.1	0.1	0.1	0.15	0.15	0.15	0.2	0.2	0.2			
f [mm/rev]	0.1	0.1	0.1	0.2	0.2	0.2	0.3	0.3	0.3	0.4	0.4	0.4			
	Hole diameter depending on regime												min	max	
PomC	6.641	6.695	6.712	6.662	6.671	6.671	6.664	6.665	6.653	6.731	6.746	6.721	6.641	6.746	0.105
PEHD 1000	6.505	6.55	6.584	6.565	6.562	6.568	6.601	6.613	6.607	6.566	6.615	6.644	6.505	6.644	0.139
PA6	6.604	6.64	6.654	6.632	6.609	6.631	6.633	6.629	6.62	6.667	6.675	6.661	6.604	6.675	0.071
M960	6.712	6.717	6.707	6.676	6.686	6.693	6.675	6.675	6.681	6.724	6.726	6.729	6.675	6.729	0.054
M980	6.697	6.709	6.697	6.693	6.694	6.702	6.705	6.693	6.701	6.644	6.697	6.688	6.644	6.709	0.065
M700	6.694	6.705	6.689	6.685	6.682	6.673	6.685	6.685	6.683	6.697	6.691	6.698	6.673	6.705	0.032

Depending on the properties of the material, we find that the diameter of the hole is smaller by 0.1–0.3 mm, as it is 1.5–4.4% of the measured diameter of the tool holder assembly with the drill used (% of the diameter of the drill).

In extremely elastic materials, such as PEHD1000, the biggest variances are found in relation to tool diameter. The diameter of the holes machined in the rigid SikaBlock materials have a slight difference compared to the diameter of the tool used.

In the case of POM-C, PEHD1000, and PA6 materials, the cutting regime influences the diameter of the processed hole. Depending on the cutting regime chosen, the diameter of the hole has a variation of 0.139–0.071 mm (Δ diameter from Table 7).

In SikaBlock materials (M960, M980, M700), the chipping regime does not have a major influence on the hole diameter. The holes obtained have a variation of 0.032–0.065 mm (Δ diameter from Table 7).

## 6. Determination of the Influence of the Chipping Regime on the Precision of the Hole—With Specific Reference to the Deviation from Cylindricality, or the Resulting Diameter after Processing

The feed has a greater impact on the cylindricity of the hole than cutting speed. According to the graph, the cylindricity of the processed holes grows noticeably at cutting regimes with an advance of 0.4 mm/revolution. According to the results of the trials, there is little to no significant impact of the advancement on the cylindricity using the PEHD1000 material (Figure 10). After the investigation, cutting parameters 8 (v = 25 m/min, s = 0.3 mm/rev), where the cylindricity is the least, can be found. We achieve the hole with the best degree of precision using the aforementioned regime.

Depending on the cutting regime and the processed material, we find a difference of 0.241 mm between the minimum and largest diameter. In the case of the ϕ6.8 mm hole, this difference positions the hole in quite different tolerance fields.

Regardless of the cutting parameters applied, Table 9 displays the accuracy classes based on the minimum and maximum values of the hole.

We discover that the cutting regime with an advance of 0.4 mm/rev (0.2 mm/tooth) has detrimental effects on the processing precision after analyzing the data from Table 10 and Table 11. We draw the conclusion from the experiment that raising the feed has a substantial impact on the machining precision.

The holes are in more severe precision classes when the advance of 0.4 mm/rev (0.2 mm/tooth) is removed from the analysis of the data. The holes are one class more precise for materials such as POM-C, PEHD1000, PA6, M700, and M960. The holes are two classes more precise for the M980 material.

By analyzing the measured data, the cutting regime may be selected for each material to produce holes with an IT6–IT8 precision level, which even permits the insertion of pins. The main benefit is that the technology can do away with reaming following drilling (Table 11).

## 7. Discussions

Following the running of the experiments according to the varied parameters, we obtained different values of the hole diameter, respective to its precision. The variation is not linear over the studied interval. In each range studied, we were able to find a cutting regime that we can recommend, obtaining the highest hole accuracy.

When the advance of 0.4 mm/rev (0.2 mm/tooth) is taken out of the data analysis, the holes are in more severe precision classes. When it comes to materials like POM-C, PEHD1000, PA6, M700, and M960, the holes are one class more exact. In the M980 material, the holes are two classes more accurate.

In the case of Polyurethane type materials (commercial name SikaBlock), the chipping regime that ensures the best precision of the hole is in the case of the advance of 0.15 mm/tooth. It seems that at this feed, we have the best chip removal conditions. In contrast, the precision of the holes is diminished for bigger (0.2 mm/tooth) or smaller (0.05–0.1 mm/tooth) advances, in comparison to the 0.15 mm/tooth advance.

In the case of polyurethane-type materials (brand names SikaBlock M700, M960, and M980), it can be said that the cutting speed has no effect on the processing precision between 12 and 37 m/min.

Instead, we discovered that the machining precision is affected by the cutting speed when it comes to the group of materials made of polyacetal, polyethylene, and polyamide (POM-C, PEHD100, and PA6). Cutting regimes (cutting speed connected with feed) can be found for many materials that result in excellent processing precision.

In the case of the POM-C material, following the experiments, we found that it is helpful to use reduced cutting speeds of 12 m/min to obtain high precision holes. A major influence of the feed on the hole accuracy cannot be found in the range of 0.05–0.2 mm/tooth.

When processing PA6 material, it is helpful to use high cutting speeds and low feeds. The highest accuracy of the hole was obtained in the case of the cutting regime defined by the cutting speed of 37 m/min, respective to advances of 0.05–0.1 mm/tooth.

In the case of the PEHD1000 material, the high cutting speed ensures a high processing precision. On the contrary, too little feed per tooth 0.05 mm/tooth, respective to a large feed per tooth 0.2 mm/tooth, has a negative influence on the accuracy of the hole. According to the conducted experiment, the most correct holes were obtained using the cutting regime defined by the cutting speed of 37 m/min, respective to the advance per tooth of 0.1–0.2 mm/tooth.

Following the analysis of the dimensions of the holes, we find that for each material studied, there is at least one cutting regime where the processed hole will be in the IT6–IT8 precision classes.

This achieved precision (IT6–IT8) is important from several points of view.

Holes in the IT6–IT8 precision class can only be machined using the helical drill. Reaming technology no longer requires the use of a reamer.

## 8. Conclusions

By drilling with a twist drill using the proper cutting regime, we machine holes for pins. Manufacturing costs are considerably reduced by eliminating the reaming operation.

Based on the experimental data collected and analyzed, we were able to set up the contraction value of the hole diameter compared to the diameter of the drill bit used in processing.

Based on the findings, we can formulate the following recommendation about the choice of drill diameter so that after the drilling process, the diameter of the hole is the designed one.

For 6–7 mm holes, depending on the material of the semi-finished product processed to obtain the desired diameter and compared to the diameter of the designed hole, we will use drills with a larger diameter:+0.1 mm in the case of POM-C material+0.1 mm in the case of SikaBlock M700, M960 materials+0.2 mm in the case of PA6 material+0.2 mm in the case of SikaBlock M980 material+0.3 mm in the case of PEHD 1000 material

Based on the statistical information obtained, 4 values out of a total of 288 were found to be outliers, or 1.38% of the data. A proportion of 1.38% of a batch is acceptable from a manufacturing perspective.

The R-Squared statistic shows that the model as fitted explains 91.9338% of the variability in Circularity 1 and 93.7764% of the variability in Circularity 2. The correlation coefficient equals −0.958821 for Circularity 1 and −0.968382 for Circularity 2, showing a strong relationship between the variables. The standard error of the estimate shows the standard deviation of the residuals to be 1.13177 for Circularity 1 and 0.964281 for Circularity 2.

Further research directions could be:Analyzing the experimental data through the prism of evaluating the random nature of the data and eliminating outliers from the samples.Expanding the research also on the hole diameters specific to M5, M6, M10, M12, M14 threads, threads often encountered in the parts manufactured from the studied materials.Identification of the influence of the hole diameter on the precision and quality of the processed thread.

## Figures and Tables

**Figure 1 polymers-15-00347-f001:**
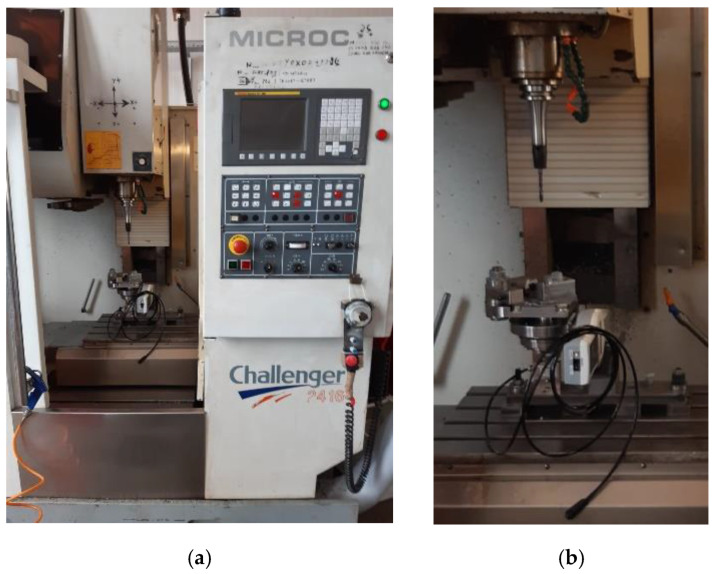
Challenger 2418 CNC machine (**a**) and specimen fixture and cutting tool (**b**).

**Figure 2 polymers-15-00347-f002:**
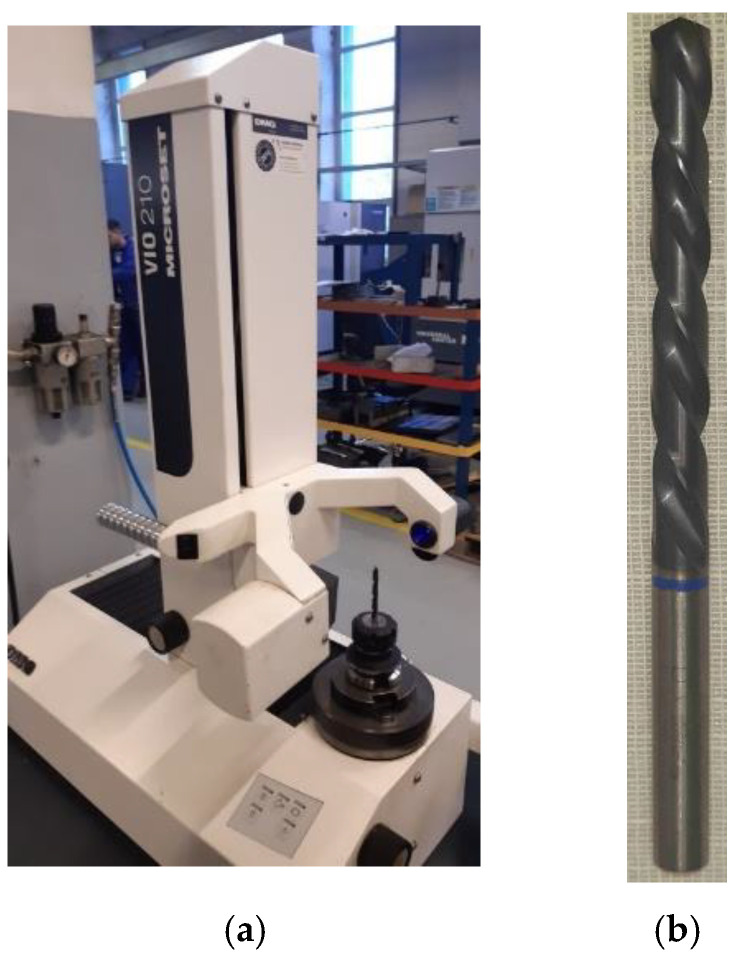
VIO 210 MICROSET cutting tool measuring machine manufactured by DMG (**a**) and the twist drill (**b**).

**Figure 3 polymers-15-00347-f003:**
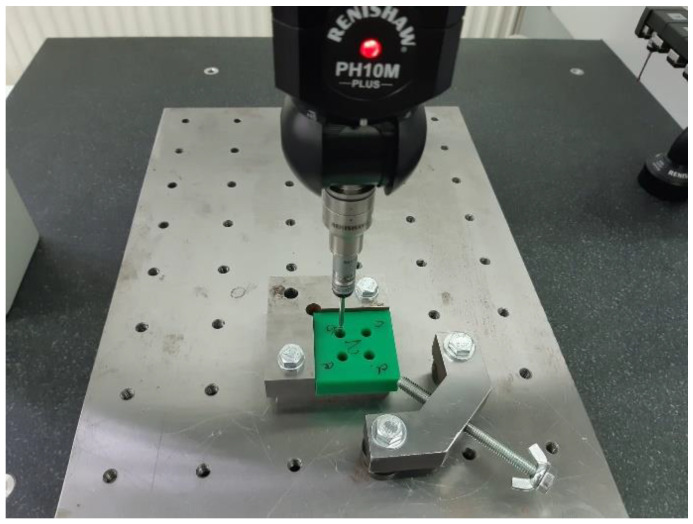
Four-hole specimen, specimen holder, and RENISHAW PH10M measuring head.

**Figure 4 polymers-15-00347-f004:**
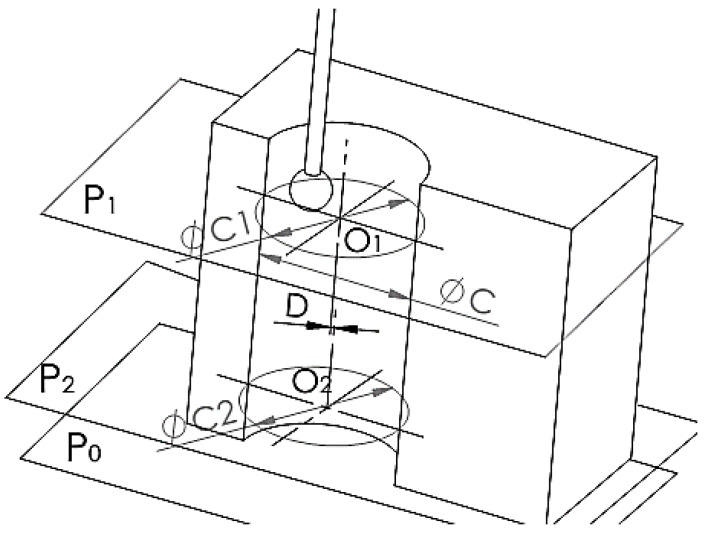
Measured elements.

**Figure 5 polymers-15-00347-f005:**
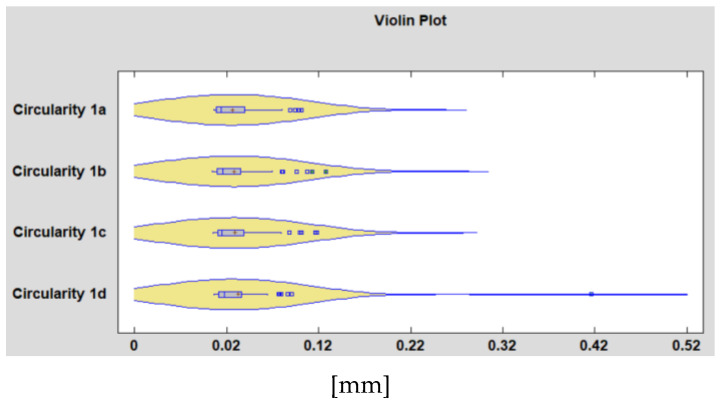
Comparative violin plot showing the measured values for the four holes on the sample’s P1 measurement plane’s deviation from circularity.

**Figure 6 polymers-15-00347-f006:**
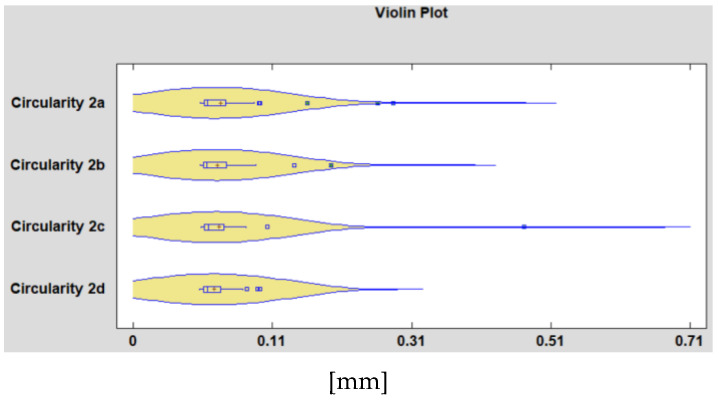
Comparative violin plot showing the measured values for the four holes on the sample’s P2 measurement plane’s deviation from circularity.

**Figure 7 polymers-15-00347-f007:**
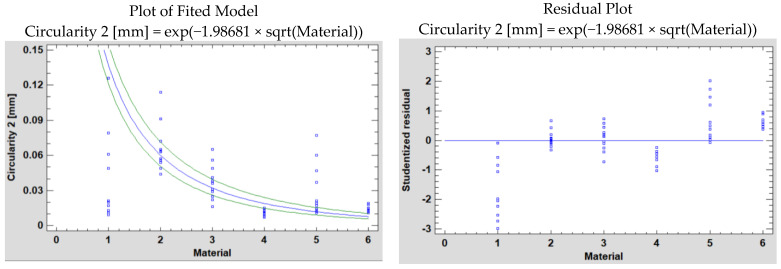
Plot of fitted model and residual for circularity 2.

**Figure 8 polymers-15-00347-f008:**
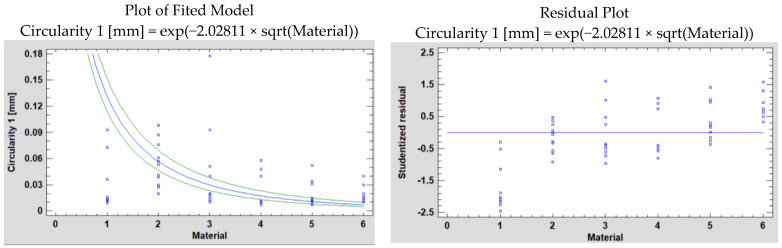
Plot of fitted model and residual plot for circularity 1 using the Logarithmic-Y square root-X model of the simple regression.

**Figure 9 polymers-15-00347-f009:**
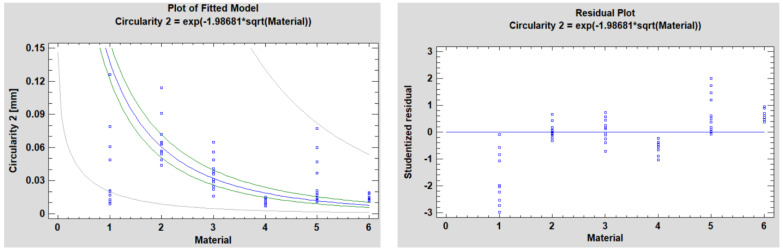
Plot of fitted model and residual plot for circularity 2 using the Logarithmic-Y square root-X model of the simple regression.

**Figure 10 polymers-15-00347-f010:**
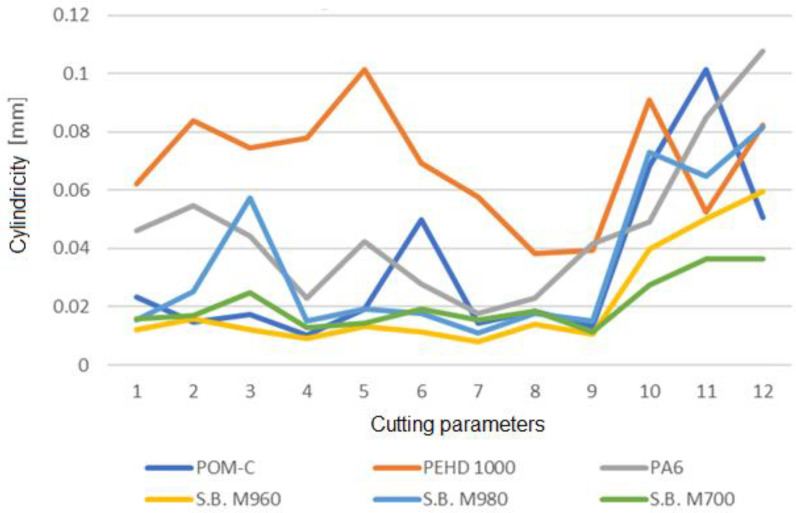
Cylindricity according to material and cutting parameters.

**Table 1 polymers-15-00347-t001:** The mechanical characteristics of the materials by the Quality Certificates issued for the purchase of semi-finished products.

	POM-C	PEHD 1000	PA6	SIKA BLOCK		
				M960	M980	M700
Density [g/cm^3^]	1.41	0.93	1.14	1.2	1.35	0.7
Shore hardness [scale D]	81	51	83	78	86	66
Flexural strength [MPa]	91	27	130	80	145	26
E-Modulus [MPa]	2800	750	3300	2200	4000	1000
Compressive strength [MPa]	68	65	68	70	120	25
Impact resistance [kJ/m^2^]	6	No Break	3	30	35	7
Heat distortion temperature [°C]	110	135	220	80	85	90
Linear thermal expansion coefficient αT [K^−1^]	110 × 10^−6^	200 × 10^−6^	100 × 10^−6^	85 × 10^−6^	60 × 10^−6^	55 × 10^−6^
Tensile strength [MPa]	67	17	85	No data	No data	No data

**Table 2 polymers-15-00347-t002:** Challenger 2418 CNC machine features.

	Unit	Value
Spindle speed	[rpm]	10,000
Spindle Power (continue/peak)	[kW]	5.5/7.5
Feed rates, maximum cutting axis x/y/z	[m/min]	10/10/10
Positioning accuracy	[mm]	+/−0.005
Repeatability	[mm]	+/−0.004

**Table 3 polymers-15-00347-t003:** Holes with manufacturing errors.

Material	Cutting Speed [m/min]	Feed [mm/rev]	Circularity [mm]	Average Sample Circularity [mm]	Hole
3	37	0.4	0.416	0.027	1d
1	37	0.3	0.472	0.025	2c
2	37	0.2	0.103		2c
2	12	0.1	0.103		2c

**Table 4 polymers-15-00347-t004:** Comparison of Alternative Models.

	Measurement Plan 1		Measurement Plan 2	
Model	Correlation	R-Squared	Correlation	R-Squared
Logarithmic-Y square root-X	−0.9588	91.93%	−0.9684	93.78%
Log probit	−0.9058	82.05%	−0.9201	84.66%
Exponential	−0.9042	81.76%	−0.9209	84.80%
Squared-Y	0.2636	6.95%	0.3134	9.82%
Double squared	0.1723	2.97%	0.1943	3.78%

**Table 5 polymers-15-00347-t005:** Coefficients.

	Least Squares	Standard	T	
Parameter	Estimate	Error	Statistic	*p*-Value
Slope P1	−2.02811	0.071295	−28.4468	0.0000
Slope P2	−1.98681	0.060744	−32.708	0.0000

**Table 6 polymers-15-00347-t006:** Analysis of Variance.

Source	Measuring Plane	Sum of Squares	Df	Mean Square	F-Ratio	*p*-Value
Model	P1	1036.53	1	1036.53	809.22	0.0000
	P2	994.753		994.753	1069.81	
Residual	P1	90.9446	71	1.28091		
	P2	66.0185		0.929838		
Total	P1	1127.48	72			
	P2	1060.77				

**Table 8 polymers-15-00347-t008:** Recommendations about the choice of the diameter of the drill to obtain the hole with the prescribed (designed) diameter.

	Material	Diameter [mm]		Compared to 6.824 [mm]			To Obtain the 6.8 mm Hole, Choose a Tool with a Larger Diameter	Drill Diameter to Be Fixed in the Tool Holder [mm]	Diameter Measured Drill along with Tool Holder [mm]
		Min	Max	Min	Max	Average			
1.	POM-C	6.641	6.746	0.183	0.078	0.1305	+0.1 mm	6.7 + 0.1 = 6.8 mm	6.824 + 0.1 = 6.92 mm
2.	PEHD 1000	6.505	6.644	0.319	0.18	0.2495	+0.3 mm	6.7 + 0.3 = 7.0 mm	6.824 + 0.3 = 7.12 mm
3.	PA6	6.604	6.675	0.22	0.149	0.1845	+0.2 mm	6.7 + 0.2 = 6.9 mm	6.824 + 0.2 = 7.02 mm
4.	M960	6.675	6.729	0.149	0.095	0.122	+0.1 mm	6.7 + 0.1 = 6.8 mm	6.824 + 0.1 = 6.92 mm
5.	M980	6.644	6.709	0.18	0.115	0.1475	+0.2 mm	6.7 + 0.2 = 6.9 mm	6.824 + 0.2 = 7.02 mm
6.	M700	6.673	6.705	0.151	0.119	0.135	+0.1 mm	6.7 + 0.1 = 6.8 mm	6.824 + 0.1 = 6.92 mm

**Table 9 polymers-15-00347-t009:** Accuracy classes according to the minimum and maximum values of the hole regardless of the cutting regime.

No.	Material	Minimum Diameter [mm]	Maximum Diameter [mm]	Tolerance [mm]	Class
	POM-C	6.641	6.746	0.105	IT12
	PEHD 1000	6.505	6.644	0.139	IT13
	PA6	6.604	6.675	0.071	IT11
	M960	6.675	6.729	0.054	IT11
	M980	6.644	6.709	0.065	IT11
	M700	6.673	6.705	0.032	IT10

**Table 10 polymers-15-00347-t010:** Hole accuracy classes after removing from the analysis the 0.4 mm/rev (0.2 mm/tooth) feed regime.

No.	Material	Minimum Diameter [mm]	Maximum Diameter [mm]	Tolerance [mm]	Class
	POM-C	6.641	6.712	0.071	IT11
	PEHD 1000	6.505	6.613	0.108	IT12
	PA6	6.604	6.654	0.05	IT10
	M960	6.675	6.717	0.042	IT10
	M980	6.693	6.709	0.016	IT8
	M700	6.673	6.705	0.032	IT9

**Table 11 polymers-15-00347-t011:** Centralizing table of hole tolerances depending on the cutting regime.

Hole Tolerance Depending on Cutting Parameters												
No. cutting regime	1	2	3	4	5	6	7	8	9	10	11	12
v [m/min]	12	25	37	12	25	37	12	25	37	12	25	37
s [mm/tooth]	0.05	0.05	0.05	0.1	0.1	0.1	0.15	0.15	0.15	0.2	0.2	0.2
f [mm/rev]	0.1	0.1	0.1	0.2	0.2	0.2	0.3	0.3	0.3	0.4	0.4	0.4
POM-C Average of measured cylindricity	0.0235	0.0149	0.0175	0.0104	0.0193	0.05	0.0142	0.0183	0.0133	0.0683	0.1013	0.0505
Min [mm]	0.019	0.009	0.01	0.007	0.012	0.013	0.007	0.011	0.007	0.055	0.089	0.038
Max [mm]	0.033	0.023	0.026	0.014	0.031	0.15	0.046	0.029	0.026	0.077	0.141	0.072
	0.014	0.014	0.016	0.007	0.019	0.137	0.039	0.018	0.019	0.022	0.052	0.034
IT	7	7	8	6	8	12	10	8	8	8	10	9
PEHD 1000Average of measured cylindricity	0.0623	0.084	0.0745	0.0779	0.1015	0.0691	0.0578	0.0384	0.0395	0.091	0.0525	0.0823
Min [mm	0.043	0.057	0.04	0.03	0.053	0.035	0.031	0.025	0.024	0.055	0.023	0.048
Max [mm]	0.095	0.112	0.128	0.185	0.24	0.134	0.173	0.078	0.057	0.125	0.092	0.115
	0.052	0.055	0.088	0.155	0.187	0.099	0.142	0.053	0.033	0.07	0.069	0.067
IT	10	10	11	13	13	12	12	10	9	11	11	11
PA6 Average of measured cylindricity	0.0462	0.0547	0.0441	0.0229	0.0424	0.0277	0.0176	0.0231	0.0418	0.0491	0.085	0.1078
Min [mm]	0.015	0.027	0.027	0.007	0.032	0.016	0.01	0.009	0.022	0.023	0.055	0.077
Max [mm]	0.113	0.092	0.061	0.05	0.052	0.038	0.03	0.105	0.062	0.078	0.104	0.188
	0.098	0.065	0.034	0.043	0.02	0.022	0.02	0.096	0.04	0.055	0.049	0.111
IT	12	11	9	10	8	8	8	12	10	10	10	12
S.B. M960 Average of measured cylindricity	0.012	0.0159	0.0123	0.0092	0.0131	0.0115	0.0082	0.0139	0.0107	0.0396	0.0502	0.0597
Min [mm]	0.007	0.009	0.008	0.006	0.009	0.006	0.005	0.01	0.007	0.034	0.04	0.046
Max [mm]	0.017	0.022	0.019	0.015	0.019	0.015	0.012	0.019	0.016	0.044	0.058	0.07
	0.01	0.013	0.011	0.009	0.01	0.009	0.007	0.009	0.009	0.01	0.018	0.024
IT	7	7	7	6	7	6	6	6	6	7	8	9
S.B. M980 Average of measured cylindricity	0.0156	0.0251	0.0572	0.0153	0.0194	0.0178	0.0111	0.0179	0.015	0.0731	0.0648	0.0815
Min [mm]	0.012	0.016	0.034	0.005	0.008	0.007	0.006	0.006	0.008	0.054	0.055	0.054
Max [mm]	0.023	0.037	0.085	0.11	0.037	0.025	0.017	0.029	0.028	0.093	0.077	0.126
	0.011	0.021	0.051	0.105	0.029	0.018	0.011	0.023	0.02	0.039	0.022	0.072
IT	7	8	10	12	9	8	7	9	8	10	8	11
S.B. M700 Average of measured cylindricity	0.0158	0.0171	0.0248	0.0129	0.0142	0.0191	0.0155	0.0185	0.0115	0.0273	0.0363	0.0363
Min [mm]	0.009	0.012	0.016	0.008	0.006	0.01	0.008	0.011	0.008	0.012	0.021	0.028
Max [mm]	0.026	0.023	0.035	0.019	0.018	0.052	0.021	0.029	0.016	0.04	0.05	0.044
	0.017	0.011	0.019	0.011	0.012	0.042	0.013	0.018	0.008	0.028	0.029	0.016
IT	8	7	8	7	7	10	7	8	6	9	9	8

## Data Availability

Not applicable.

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
