# Peer review of "Determination of Processing Precision of Hole in Industrial Plastic Materials"

_polymers, 2023, doi:10.3390/polym15020347_

Round 1
Reviewer 1 Report
In introduction section, the literatures review part should summarise the finding from previous researchers not simply list the work they have done. The description of ref.3-5, ref. 7-10, ref.15-17, ref.20-23 is too simplified, which can not obtain any reviewed information. The presentation of literature review cannot clearly see the knowledge gap.
In your design of experiment, it aims to find the influences in hole’s diameter, circularity and circularity by different processing parameters. Each type material should be analysed separately. There are 12*4 holes for each type materials, 7 measurements for each holes, so the total data can obtained for each type should be 336. In Figure 5-8, it looks like you mixed 6 type material together and separated the 4 hole result, also each hole only presented 1 measurement. This is not fit the logic from my point of view. Though there is quite detailed data analysis following, it cannot lead the scientific conclusion. In overall, it cannot convince the reader for the research outcome. The author should make a major revision in this part.
The figure is directly jumped from figure 11 to figure 17, there is no figure 12-16.
Reviewer 2 Report
The work presented by the authors starts from an interesting premise but needs a thorough revision. The structure of the work must be well reorganized and the discussion of the results should be deepened. The work will be reconsider after a major revision.
-The introduction is adequate, although it could be improved with some references from relevant authors.
-Section 2.1 should not be included in section 2. It is an independent section and it would be more appropriate to include it at the end of the introduction justifying why these objectives are set.
-It would be convenient to include a drawing or photograph of the drill bit used with the geometric parameters.
-The histograms in section 4 do not differentiate between the 6 materials used, so they should be in a section where the diameter is estimated based on the cutting parameters.
-Table 4 is redundant with Table 3 and does not offer relevant information.
-Units are missing from many tables, which is quite relevant since the paper has an important weight related to metrology, as well as including errors in the use of the international system of units. Tolerances, as a rule, must be indicated in microns if they are less than a millimeter. The advance of the tool must be indicated in mm/rev and not in mm/rot. It is also not appropriate to use the units mm/tooth in a double-edged drill. This parameter is usually indicated for multi-edge tools such as milling cutters.
-The use of the expresion "cutting mode" to refer to the cutting parameters is not adequate. It is better to use the expressions "cuttig parameters" or "cutting conditions"
-The conclusions must be much more concise, and not so extensive, differentiating which are the main contributions of the work after the study.
-In some sections I understand that the four holes made for each pair of cutting conditions are being compared, while in other sections I understand that the average values of the four holes are used. It's not explained properly and it's quite confusing to follow the reasoning.
-Tool wear has not been considered and is a relevant parameter in hole cylindricity, as well as tool overhang.
-In line 299 I don't understand where the mentioned values come from.
Round 2
Reviewer 1 Report
The corrections are addressed based on previous review.
Several addtional corrections need be addressed in this revised version:
1. In table 7, the feed rate shoud use mm/rev which is relate to the defined variation levels in section 2.
2. In line 301, "table 5: should be table 4 which is make sence.
3. Further proofread need to carry out to minimise the typos and figure table numbers mismatch.
Addtional suggestion, from table 3 the holes with manufacturing errors occured 3 times in plane 2, figure 7 should plot the data for circularity 2 which can better present the anomalies.
Reviewer 2 Report
All the changes have been addressed correctly.
Author Response
Thank you and we appreciate your precious time and invaluable comments on our manuscript.